# Neutrophils and Influenza: A Thin Line between Helpful and Harmful

**DOI:** 10.3390/vaccines9060597

**Published:** 2021-06-04

**Authors:** Sneha T. George, Jonathan Lai, Julia Ma, Hannah D. Stacey, Matthew S. Miller, Caitlin E. Mullarkey

**Affiliations:** 1Faculty of Health Sciences, McMaster University, Hamilton, ON L8S 4L8, Canada; georgs16@mcmaster.ca (S.T.G.); maj100@mcmaster.ca (J.M.); 2Faculty of Science, McMaster University, Hamilton, ON L8S 4L8, Canada; laij32@mcmaster.ca; 3Michael G. DeGroote Institute for Infectious Disease Research, McMaster University, Hamilton, ON L8S 4L8, Canada; staceyhd@mcmaster.ca (H.D.S.); mmiller@mcmaster.ca (M.S.M.); 4McMaster Immunology Research Centre, McMaster University, Hamilton, ON L8S 4L8, Canada; 5Department of Biochemistry and Biomedical Sciences, McMaster University, Hamilton, ON L8S 4L8, Canada

**Keywords:** influenza virus, neutrophils, NETs, ADCP, universal influenza virus vaccines, broadly neutralizing antibodies, Fc-mediated effector functions

## Abstract

Influenza viruses are one of the most prevalent respiratory pathogens known to humans and pose a significant threat to global public health each year. Annual influenza epidemics are responsible for 3–5 million infections worldwide and approximately 500,000 deaths. Presently, yearly vaccinations represent the most effective means of combating these viruses. In humans, influenza viruses infect respiratory epithelial cells and typically cause localized infections of mild to moderate severity. Neutrophils are the first innate cells to be recruited to the site of the infection and possess a wide range of effector functions to eliminate viruses. Some well-described effector functions include phagocytosis, degranulation, the production of reactive oxygen species (ROS), and the formation of neutrophil extracellular traps (NETs). However, while these mechanisms can promote infection resolution, they can also contribute to the pathology of severe disease. Thus, the role of neutrophils in influenza viral infection is nuanced, and the threshold at which protective functions give way to immunopathology is not well understood. Moreover, notable differences between human and murine neutrophils underscore the need to exercise caution when applying murine findings to human physiology. This review aims to provide an overview of neutrophil characteristics, their classic effector functions, as well as more recently described antibody-mediated effector functions. Finally, we discuss the controversial role these cells play in the context of influenza virus infections and how our knowledge of this cell type can be leveraged in the design of universal influenza virus vaccines.

## 1. Introduction

Neutrophils are among the first innate cells recruited to sites of infection and they possess a repertoire of essential effector mechanisms that help control the spread of pathogens, including viruses. It has become increasingly evident in the last decade that neutrophils perform a larger role in modulating the immune response to viral infections than previously thought [1]. Beyond their well-described innate effector functions, a growing body of evidence points to the ability of neutrophils to shape the adaptive immune response. Some of these responses include promoting T-cell migration, impacting the differentiation of T cell subsets, modulating dendritic cell function, and executing Fc-mediated effector functions.

Neutrophils are polymorphonuclear granulocytes that possess a segmented, multilobular nucleus [2]. Neutrophils, comprising 50–70% of all circulating leukocytes, are the most abundant white blood cell population in the blood [3]. This leukocyte population is produced and undergoes maturation in the bone marrow via granulopoiesis, a process regulated by a number of growth factors and cytokines, most notably granulocyte colony-stimulating factor (G-CSF) [4]. Mobilization of neutrophils from the bone marrow into circulation is similarly dependent on G-CSF, along with the activation of chemokine receptor CXCR2 [5,6].

Circulating mature neutrophils are quiescent and are traditionally accepted to have a lifespan shorter than one day under homeostatic conditions [7,8]. As a consequence of their short-lived nature, large reserves of neutrophils are kept in the bone marrow where in the case of an infection, neutrophils can be subsequently mobilized and released [8]. This dogma of neutrophil lifespan stems from early studies using cell transfer or radioactive tracer techniques [7,9,10]. However, more recent labelling of human neutrophils using in vivo models suggest this value is likely an underestimate and that the average circulatory neutrophil lifespan is on the order of days. More specifically, Pillay and colleagues used ^2^H_2_0 labelling of neutrophils from five healthy human volunteers and estimated the average circulatory lifespan of neutrophils to be 5.4 days [11]. In the absence of activation factors, aged and non-apoptotic neutrophils return to the bone marrow through a mechanism facilitated by the CXCR4/CXCL12 chemoattractant pathway and undergo apoptosis and clearance by resident macrophages [8,12,13]. Senescent neutrophils that undergo apoptosis while in circulation are removed by macrophages in the liver or spleen [8].

In the presence of inflammatory signals, neutrophils are recruited to sites of inflammation. Neutrophils extravasate from circulation to sites of inflammation/injury via a well-defined and controlled adhesion cascade, consisting of neutrophil capture, rolling, arrest, crawling, and diapedesis, which has been reviewed in detail by Schnoor and colleagues [14]. The high degree of flexibility and deformation required for neutrophil transmigration has largely been attributed to its characteristic segmented nucleus. Nonetheless, this long-standing view is not well-supported by empirical data and has been challenged by recent reviews that highlight the significance of nuclear envelope proteins, such as lamins and lamin B receptors in conferring nuclear flexibility instead [15]. Once neutrophils have migrated to the affected tissues, they mount an immune response through a diverse array of mechanisms. Direct classical effector functions that promote pathogen clearance include, but are not limited to, phagocytosis, release of antimicrobial granular constituents, and reactive oxygen species (ROS) production; indirect effector functions include the secretion of pro-inflammatory cytokines [16].

Here, we review the classical neutrophil effector functions and summarize recent advancements in the less well-understood antibody-dependent effector functions of neutrophils. We then focus on the involvement of neutrophils in influenza A virus (IAV) infections to evaluate the beneficial and potentially detrimental role of these cells in influenza disease progression. Finally, we conclude with a discussion of the possible roles neutrophils may play in the development of a universal influenza vaccine.

## 2. Murine vs. Human Neutrophils 

Murine models are incredibly useful as tractable systems to probe the complexities of the immune system, permitting the evaluation of phenomena that would not be possible either in vitro or in humans. However, there are important differences between human and murine physiology, which must be considered before translating findings between these species. One of the primary differences between mouse and human neutrophils is their relative abundance in circulation. Neutrophils comprise only 10–25% of peripheral murine leukocytes, which is two to seven times less than in humans [3,17]. As a result, experimental procedures resort to the isolation of murine neutrophils from the bone marrow, as they are abundant in this compartment. While bone marrow neutrophils of mice are an attractive alternative, they demonstrate a lower affinity to the *Escherichia coli* (*E. coli*) derived N-formyl peptide, formyl methionyl-leucyl-phenylalanine (fMLF) as compared to human and murine peripheral blood neutrophils [18,19]. fMLF is widely used in studies involving neutrophil activation, as it is a crucial inducer of bactericidal neutrophil functions such as superoxide production and degranulation. Hasenberg et al. also express caution about the difference in maturity between bone marrow and circulating neutrophils, as peripheral neutrophils have responded to functional mobilization stimuli such as G-CSF [20]. Along similar lines, murine neutrophils also lack defensins in their azurophilic granules [21]. These cationic peptides have been shown to participate in neutrophil antiviral activity by inhibiting viral protein synthesis in IAV-infected cells and inducing the release of the chemoattractant, IL-8 [22]. Finally, human and murine neutrophils also differ in their Fc receptor (FcR) expression patterns. Murine neutrophils express all FcγR subtypes, of which, FcγRI, FcγRIII, and FcγRIV are activating receptors and FcγRIIB is the sole inhibitory receptor; although levels of FcγRI are reported to be low [23]. In contrast, human neutrophils constitutively express the activating receptors FcγRIIA and FcγRIIIB, in addition to the inhibitory receptor FcγRIIB; expression of FcγRI expression is inducible [23,24]. Furthermore, FcαRI, an activating receptor which binds immunoglobulin A (IgA), is highly expressed on human neutrophils. However, mice have no equivalent orthologue [23]. FcRs are crucial to the antibody-mediated effector functions of neutrophils and thus, extending conclusions observed in murine systems to that of humans should be drawn with care and caution.

The use of transgenic mice which express the absent FcR equivalents is one potential solution to studying human neutrophil antibody-mediated immune responses. Over the past two decades, many mouse models have been developed in an attempt to recapitulate human conditions. Relevant models include hFcαRI^tg^ [25,26,27]; hFcγRI^tg^ [28], hFcγRIIA^tg^ [29], hFcγRIIB^tg^ [30], and hFcγRIIIB^tg^ [31]. Nevertheless, some transgenic mouse models are limited in ways and can only partially mimic human conditions. For example, in hFcγRI^tg^ mice, FcγRI is constitutively expressed, although as noted earlier, the physiological expression of this FcR in human neutrophils is inducible. [32]. Similarly, in mice with the FcαRI transgene, despite producing the anticipated expression pattern in neutrophils, the FcαRI expression levels in monocytes were only reflected in a subpopulation of peripheral monocytes [25,27].

## 3. Classical Neutrophil Effector Functions

### 3.1. Phagocytosis 

Activated neutrophils possess a number of mechanisms to eliminate pathogens. The most well-studied mechanism is the process of phagocytosis, where receptor-mediated endocytosis leads to the engulfment of a pathogen [33] (Figure 1a). Neutrophil phagocytosis specifically involves FcγR—FcγRIIA (CD32) and FcγRIIB (CD16) and complement receptors CR1 and CR3 [34]. After engulfment of a foreign pathogen or IgG-opsonized particles, a phagosome forms around the particle. The mature phagosome mediates pathogen clearance using a variety of hydrolytic enzymes and NADPH oxidase subunits, which can also result in collateral damage to surrounding cells [35].

### 3.2. Degranulation

Neutrophils can also eliminate pathogens through the extracellular release of potent cytotoxic granules (Figure 1b). There are three types of neutrophilic granules: primary (azurophilic), secondary (specific) and tertiary (gelatinase) [34,35]. Secondary and tertiary granules are also able to induce the NADPH oxidase pathway, producing ROS to further create a toxic environment [35]. The process of degranulation releases primary and secondary granules and helps to confer an inhospitable antimicrobial microenvironment at the inflammatory site [36].

In addition to the direct effects of degranulation, recent work has also uncovered its role in shaping the adaptive immune response as well. Minns and colleagues demonstrated that the antimicrobial peptide, cathelicidin, released in the presence of TGF-β1 promotes the differentiation of CD4+ T cells to a Th17 phenotype [1]. Th17 CD4+ T cells are characterized by their ability to produce IL-17 and IL-6, which enhance neutrophil responses at the site of infection. Th17 responses were suppressed in cathelicidin-deficient mice during the process of inflammation, but this phenomenon was not observed at steady-state [1]. Taken together, these data identified cathelicidin as an important Th17 differentiation factor and defined further mechanisms by which neutrophils influence adaptive immune responses [1].

### 3.3. Reactive Oxygen Species (ROS)

ROS are chemically reactive metabolites of oxygen that come from the by-products of aerobic metabolism [37,38] (Figure 1c). Neutrophils synthesize a variety of ROS to aid in pathogen killing and clearance. ROS production is achieved chiefly through the activation of NADPH oxidase, a multicomponent enzyme that requires electrons to reduce oxygen to superoxide. The superoxide ion can then spontaneously dismutate to hydrogen peroxide; both of these primary ROS compounds can then lead to the formation of secondary ROS through the neutrophil myeloperoxidase (MPO) enzyme [37]. Furthermore, neutrophils can release these compounds into the extracellular space or into the phagosome after phagocytosis. ROS production in the absence of phagocytosis can also be induced through stimulation with phorbol ester, phorbol myristate acetate (PMA), and various lectins [37]. In addition to directly damaging pathogens, ROS can also increase cytokine production and recruit other pro-inflammatory mediators to the site of infection [39].

### 3.4. Cytokine Production 

Neutrophils synthesize a variety of cytokines critical to the stimulation of innate and adaptive immune responses (Figure 1c). Cytokine production has been extensively reviewed by several other groups [34,36,40,41,42], and therefore it will only be briefly discussed below. Neutrophils express and produce a variety of pro- and anti-inflammatory cytokines (e.g., IL-6 and TNFα), chemokines (e.g., CXCL8 and CXCL10), colony-stimulation and angiogenic factors (e.g., VEGF), TNF family members (e.g., TRAIL), and growth factors (e.g., HB-EGF) [34,36,40,41]. As compared to macrophages or leukocytes, neutrophils produce lower amounts of cytokines per cell [36]. Yet, the sheer abundance of neutrophils at the site of infection can compensate for this shortcoming and therefore significantly impact the local milieu [36]. Chemokines produced by neutrophils play a role in regulating leukocyte trafficking and recruiting other cells to the site of infection, primarily other neutrophils, monocytes, dendritic cells, natural killer cells, and CD8+ T cells [41]. Neutrophils are also able to elicit a variety of antibody-dependent effector functions that link the innate and adaptive immune response together. In particular, the Fcγ receptor is upregulated in the presence of inflammatory cytokines, including IFNγ which promotes antibody-dependent cellular toxicity (ADCC) [42]. Finally, neutrophils respond to proinflammatory cytokines such as IL-1 and TNFα, which function to enhance neutrophil effector functions such as the production of ROS [34].

## 4. Antibody Mediated Effector Functions 

### 4.1. Antibody Dependent Cellular Phagocytosis 

Neutrophils can interact with the humoral compartment through a variety of mechanisms. Antibody-dependent cellular phagocytosis (ADCP) is one such mechanism that enables clearance of pathogens, including viruses (Figure 1d). ADCP occurs when FcγRs on the surface of neutrophils engage with the Fc-region of an antibody-opsonized target, causing internalization of the target and subsequent degradation. Clustering of Fc-receptors is critical to this process as it has been shown that early signaling events are proportional to the number of engaged Fc receptors [43]. The ability of influenza-specific antibodies to induce ADCP in both human and mouse neutrophils has been demonstrated using IAV: immune complexes by our group [44]. As with other influenza-specific Fc-mediated effector functions, ADCP is impacted by epitope specificity. The influenza hemagglutinin protein (HA) contains two distinct epitope regions, which include the antigenically conserved stalk region and immunodominant variable head domain. Early research by DiLillo and colleagues demonstrated a requirement for antibody-FcγR for optimal protection by HA stalk-specific antibodies, but not antibodies that recognized the head domain [45]. Using an in vitro luminol-based assay we showed that HA-stalk specific antibodies could induce ROS production and phagocytosis in an Fc-dependent manner [44]. This was observed for both HA-stalk specific IgA and IgG [44]. These results support and expand on previous work that concluded sera from healthy human donors mediates HA-specific ADCP in vitro using a monocytic cell line [46]. Overall, these data indicate that ADCP is induced by HA-specific antibodies and help to illustrate the breadth of mechanisms elicited in the context of an IAV infection.

### 4.2. Antibody-Dependent Respiratory Burst

Neutrophils can also produce ROS in an antibody-dependent fashion through a process known as antibody-dependent respiratory burst (ADRB) (Figure 1e). ROS are generated by NADPH oxidase and as discussed previously, these can be released into the phagosome or the extracellular space [35]. The literature surrounding ADRB and pathogen clearance is largely focused on parasites such as *Plasmodium* spp., where neutrophils are thought to contribute to the clearance of blood-stage parasites. ADRB has also been examined as possible correlates of protection for candidate malaria vaccines [47,48]. In humans, Kapelski and colleagues identified CD32A as the main FcγR on human neutrophils involved in ADRB in response to IgG bound to solid-phase immobilized malarial antigens [49]. Studies by Llewellyn et al. in mice concluded that ADRB activation is FcγRIII (CD16)-mediated [50]. Similar to ADCP, ADRB is thought to require Fc-FcR interactions for full activation, further highlighting the importance of FcRs in antibody-specific mechanisms of killing pathogens. However, in the context of a viral infection, where replication occurs intracellularly, the importance of extracellular ROS production or ROS production in the absence of phagocytosis is unclear. It is possible that ADRB could be elicited by influenza-specific antibodies, but to the best of our knowledge this has not been reported in the literature. 

### 4.3. Neutrophil Extracellular Traps (NETs)

Neutrophils are capable of undergoing a unique form of cell death, NETosis, which results in the formation of neutrophil extracellular traps (NETs). NETs are composed of decondensed chromosomal DNA, histones, and granular proteins capable of trapping and eliminating extracellular microbes and were first identified by Brinkmann and colleagues in 2004 [51]. Initial in vivo work by this group suggested NETs were abundantly present in areas of acute inflammation and were critical to the innate immune response by binding and trapping foreign material [51]. Current research has highlighted two different forms of NETosis: “classical/suicidal” NETosis and “vital” NETosis; the former is characterized as a specialized form of programmed cell death and the latter allows for the neutrophil to retain functionality and viability [52] (Figure 1f). In classical NETosis, stimulation from a variety of inflammatory mediators (ex. TNF, IL-8) and activation of toll-like receptors (TLRs), cytokine stimulation, and Fc-receptors activation, cause NETs to be released via the process of NETosis [35]. On the other hand, vital NETosis also involves the release of decondensed chromatin and histones into the extracellular space yet the neutrophil is still able to continue to function. It has been suggested that the process of vital NETosis allows for neutrophils to maintain their membrane integrity, optimize capture of bacteria and viruses within the bloodstream, and utilize conventional host defense mechanisms in the absence of neutrophil lysis [53]. Yet, the contexts and relative regulation of vital versus suicidal NETosis remain incompletely understood.

Previous work has demonstrated the need to simultaneously stimulate several receptors in order to effectively trigger NETs [54]. Papayannopoulos et al. conducted a study that demonstrated the role of neutrophil elastase (NE) and MPO in the regulation of NETs. Their results indicated that neutrophil activation causes NE to be released from azurophilic granules followed by MPO, leading to chromatin decondensation [55]. MPO is also important for the production of hypochlorous acid and induction of oxidative burst to create an inhospitable environment for pathogens [35]. Overall, their data support a model whereby ROS production is initiated leading to the subsequent release of NE and MPO, which in turn go on to drive chromatin decondensation [55].

The most common chemical used in experimental studies to trigger NET formation is PMA, as it activates protein kinase C [56]. Björnsdottir and colleagues demonstrated that neutrophil PMA-triggered NET formation requires the production of ROS. Using human neutrophils from healthy donors their results showed that upon PMA stimulation, intracellular ROS production was found within granules [56]. They went on to establish that NET formation was inhibited in the presence of an MPO-inhibitor. This was a significant finding as their data confirmed that without ROS formation and the presence of MPO within intracellular granules, NET formation cannot occur [56].

NETs can also be induced in an Fc-dependent manner. Many studies have focused on the importance of Fc-FcγR and their ability to induce NET formation [57,58]. However, Aleyd and colleagues first described a role for the FcαRI receptor in the process of NETosis. Their data demonstrated that IgA-opsonized bacteria elicit NETosis and blocking the FcαRI receptor abrogated both phagocytosis and NET formation [59]. Most recently, our group showed that IgA virus immune complexes are also able to potentiate NETosis in a process that is dependent on FcαRI engagement and NADPH oxidase complex-dependent pathways. Our results also demonstrated that NETs are capable of trapping and inactivating IAV in vitro, illustrating another neutrophilic Fc-mediated effector function engaged during influenza virus infection [60]. 

Overall, the research surrounding NET formation during infection suggests that whether NETs are protective or pathogenic is highly context dependent. NETs have been shown to aid in the elimination and clearance of pathogens [61,62,63], however, they have also been implicated in the pathogenesis of autoimmune and inflammatory disorders [64,65,66]. This topic has been extensively reviewed by multiple groups in the context of virus-induced infection, demonstrating both potential benefits in disease mitigation and pitfalls associated with uncontrollable NETosis [53,67]. 

## 5. Influenza Viruses and Vaccines

Influenza viruses are one of the most common respiratory pathogens and annual global infections result in high rates of morbidity and mortality [68,69]. While influenza virus epidemics recur on a seasonal basis during the winter months, influenza virus pandemics occur stochastically but often result in more serious illness and fatalities [69]. Notably, the “Spanish influenza” pandemic that occurred between 1918–1919 claimed approximately 50 million lives, classifying it as the worst pandemic to date [70,71]. The most recent pandemic of H1N1 occurred in 2009 and with evidence of influenza pandemics dating back to ancient civilization, it is clear that influenza viruses will continue to pose a significant threat to global public health [72]. 

Influenza viruses are enveloped viruses of the family *Orthomyxoviridae*; they are composed of a host cell-derived lipid membrane and a single stranded-segmented RNA genome [69,72,73,74]. Of the different genera of *Orthomyxoviridae*, the most clinically relevant to humans are IAVs and influenza B viruses (IBVs); specific strains of IAVs and IBVs are included in seasonal influenza virus vaccine preparations. The capacity of IAVs to infect a wide range of hosts, most notably swine, poultry, waterfowl, and humans, can occasionally result in the emergence of a novel viral strain to which humans have little-to-no prior immunity. This process, termed antigenic shift, occurs through genetic reassortment when one cell becomes infected with viruses from multiple host species and can lead to the emergence of a pandemic [75]. The rest of this review will focus on IAVs as these infections are the most robustly described in experimental models and in vivo.

On the surface, IAV expresses three main proteins: hemagglutinin (HA), neuraminidase (NA), and matrix 2 (M2). The primary target cells of IAV are epithelial cells found in the respiratory tract and HA facilitates entry into these cells by binding surface sialic acid residues [69,76]. IAVs are further classified into subtypes on the basis of the antigenic properties of HA and NA [75]. There are currently 18 HA and 11 NA subtypes. The HA proteins can be further broken down into Group 1 (H1, H2, H5, H6, H8, H9, H11, H12, H13, H16) and Group 2 (H3, H4, H7, H10, H14, H15). However, the most recent H17N10 and H18N11 viruses discovered in bats are unique in that their HAs do not bind sialic acid, but rather MHC Class II [77,78,79]. Similarly, N10 and N11 lack sialidase activity, again highlighting the distinct entry and exit mechanisms of these subtypes and the plasticity of IAVs [80,81].

Presently, yearly vaccines are widely used for the prevention and control of influenza virus infections. Three classes of these seasonal vaccines are available, including inactivated, live attenuated, and recombinant HA; all are multivalent and contain the predicted influenza strains for the upcoming season [82]. At the population level, vaccines work by inducing “herd immunity,” which refers to the maintenance of a critical level of immunity in the population; the virus spread is thus stemmed when the probability of infection drops below a critical threshold [83,84]. Achieving herd immunity against IAV is essential in helping to protect vulnerable groups such as young children, older adults, or those who are not able to be vaccinated [82]. Current vaccines elicit antibodies that target the HA head domain and provide protection by preventing the binding of HA to sialic acid. However, the HA head domain rapidly acquires point mutations that render strain specific-antibodies ineffective, a process referred to as antigenic drift [69,72,82]. As a result, current seasonal vaccines require constant evaluation and reformulation [82]. Furthermore, antigenic drift increases the difficulty of predicting future strains, leading to potential vaccine mismatches. Licensed seasonal vaccines have several other limitations, including poor immunogenicity in the elderly and egg-based adaptations [82]. These topics have been extensively reviewed by others [82,85,86,87] and will not be addressed here. All of these factors limit the overall effectiveness of seasonal influenza virus vaccines.

The limitations of seasonal influenza virus vaccines have sparked intensive research into the potential development of universal influenza virus vaccines which would provide broader protection against a variety of influenza virus strains and ameliorate the need for annual re-formulation. The generation of antibodies against more conserved viral epitopes has emerged as one promising strategy in the development of universal influenza virus vaccines. The HA protein is composed of two regions, the HA head domain, and the membrane-proximal HA stalk. The stalk portion of the protein is less tolerant to mutations [88] and significant genetic similarity in this region is conserved among HAs belonging to the same group. Antibodies which bind to conserved regions of the HA stalk have been identified. Termed broadly-neutralizing antibodies (bnAbs), these antibodies are capable of neutralizing a wider breadth of IAVs, most often subtypes within a group [45,89]. In rare instances, antibodies with cross-group protection have been identified [90]. Many candidate universal influenza virus vaccines typically aim to induce high levels of these stalk-reactive bnAbs. The mechanism of protection elicited by bnAbs is distinct from HA-head binding antibodies, as bnAbs act at later stages of the viral life cycle such as membrane fusion or viral egress. We have shown that bnAbs potently induce Fc-activation in vitro, whereas antibodies that bind to the HA head domain do not [91]. Optimal protection by bnAb in vivo has been shown to rely on this Fc-receptor engagement [45,89]. In the context of broadly-reactive HA antibodies of the IgG isotype, Fc-mediated protection appears to be conferred chiefly by alveolar macrophages [92]. Leon and colleagues expanded on this initial observation by demonstrating that optimal activation of Fc-mediated effector functions by bnAbs requires two points of contact and that stalk-specific antibodies allow for the second point of contact between the receptor binding domain of HA and the sialic acid motifs on effector cells [93]. Therefore, understanding and leveraging FcγR-mediated effector functions is vital in the design of HA stalk-antibody-based universal influenza virus vaccines. As previously discussed, neutrophils undergo a variety of Fc-mediated effector functions, therefore these innate cells may contribute to vaccine-mediate protection against IAVs. The following sections outline the controversial roles of neutrophils during influenza virus infection and the subsequent implication for vaccine design.

## 6. Neutrophils and Influenza Virus Infection: A Complex Relationship

It is well known that neutrophils play a major role in the clearance and elimination of bacteria and fungi, though recent research has highlighted an important yet contentious role in viral infections as well [68]. Infection of the upper respiratory tract with IAVs leads to the robust recruitment of neutrophils and the subsequent production of proinflammatory cytokines. This is readily observed in mouse models of infection, where the number of neutrophils in the lung doubles at one day post-infection (p.i.) [94]. The primary response of both mouse and human neutrophils to IAVs has been examined in vitro by Wang and colleagues [95]. Their data indicate that GM-CSF primed human neutrophils produce IL-8 and MIP-1β in response to IAV and that viral entry is necessary for cytokine production [95]. Similarly, the authors show that murine neutrophils are also activated by IAVs through TLR7 to produce inflammatory cytokines [95]. Yet, with clear evidence of the activation of neutrophils by IAVs and the production of proinflammatory cytokines, should these results be interpreted as protective or pathogenic? Viral clearance relies on the recruitment of immune cells to the site of infection through the secretion of proinflammatory cytokines and chemokines, however, exacerbated levels of these same soluble mediators are observed in severe and fatal cases of IAV infection [96]. Thus, the abundant literature present to support a protective role for neutrophils is also countered by growing evidence of overactive neutrophil activity in severe disease and acute respiratory distress syndrome in humans [97,98,99]. The following sections discuss the complex role of neutrophils and their various effector functions in IAV infection. While certain data indicate the necessity of neutrophils in early defense against IAV infection, other results suggest a detrimental role for excessive neutrophils leading to immunopathology. Thus, the overall designation of their contributions as protective or pathogenic is nuanced and must be approached and interpreted carefully.

### 6.1. The Beneficial Roles of Neutrophils during Influenza Virus Infection

Studies suggesting a beneficial role for neutrophils in influenza virus infection typically rely on mouse models to monitor the impact of neutrophil depletion before or after infection with influenza viruses. For many years, the monoclonal antibody (mAb) RB6-8C5 was the standard for neutrophil depletion [100]. However, RB6-8C5 binds both Ly6G expressed on neutrophils, and Ly6C present on neutrophils, dendritic cells, and other populations of lymphocytes and monocytes [100]. Therefore, the results of these studies must be interpreted with the additional depletion of off-target immune cells in mind. In our analysis we focused on studies using the mAb 1A8, which targets solely Ly6G and therefore displays highly neutrophil-specific depletion [101]. In these types of neutrophil depletion experiments, parameters such as mortality rate, increased viral titer, weight loss, pulmonary inflammation as well as impaired production of cytokines and specific CD8+ T-cells are used as readouts [102,103,104]. Tate and colleagues were the first to utilize 1A8 in a mouse model of IAV infection to shed light on the impact of selective depletion of neutrophils [103]. In these studies, neutropenic mice developed severe disease characterized by higher titers of virus both in the lungs and at extrapulmonary sites [103]. Furthermore, neutrophil depleted mice experienced greater pulmonary inflammation, lung edema and respiratory dysfunction as compared to control animals [103]. Overall, these results suggest a protective role for neutrophils in the control and clearance of IAVs.

In addition to their direct effects on viral replication and spread, neutrophils have also been shown to impact the cellular response to IAVs. Previous work using systemic LPS administration in human volunteers described a subset of human neutrophils capable of suppressing T cell activation in a CD11b/CD18-dependent manner [105]. Building on these observations, Tak and colleagues specifically defined a role for neutrophil CD11b/CD18 during sublethal IAV infection in mice [106]. In these studies, CD11b ^-/-^ mice were infected and subsequently reconstituted with either wild-type or CD11b ^-/-^ neutrophils by adoptive transfer [106]. CD11b ^-/-^ mice showed significantly more weight loss starting at Day 5 (p.i) and significantly greater lung damage on Day 8 p.i as measured by bronchoalveolar lavage fluid (BAL) protein contents [106]. Depletion studies indicated that T-cells were responsible for the enhanced pathology in these mice. Transfer of WT neutrophils prior to IAV infection partially reversed the immunopathology, indicating an important suppressive role for neutrophil CD11b/CD18 in the course of IAV infection [106]. The authors postulated that the neutrophil CD11b/CD18 integrin can suppress influenza virus-induced, T-cell mediated pathology, possibly by restraining T-cell proliferation [106], although further work is required to elucidate these mechanisms.

Lim and colleagues provided further evidence for a beneficial role of neutrophils within the context of influenza-mediated T-cell responses. Again, using a mouse model of influenza infection, the authors demonstrated that neutrophil depletion prior to IAV infection in mice reduced viral clearance and the recruitment of both total CD8+ T-cells and influenza specific CD8+ T-cells [107]. Using an elegant combination of in vitro and in vivo imaging, migration assays, and granulocyte specific CXCL12 conditionally depleted knockout mice, Lim et al. went on to show that the neutrophils produce and deposit trails of the chemokine CXCL12 which are necessary for optimal CD8+ T-cell migration and localization in influenza virus-infected tissues [107]. Taken together, their data reinforce the importance of neutrophils in coordinating the adaptive response in mediating protection against IAVs.

Another pathway through which neutrophils may exert protective effects against IAVs is through the activation of the Nod-like receptor protein 3 (NLRP3) inflammasome in alveolar macrophages and the subsequent release of mature interleukin-1β (IL-1β). A growing body of evidence suggests that IL-1 signalling is critical in protection against IAVs through regulation of dendritic cell activation, enhanced priming of CD4+ T-cells, and the induction of germinal centers [108,109,110]. Moreover, reduced IL-1β production is also associated with increased mortality in influenza challenge models [111]. Peiró and colleagues established that neutrophils release mCRAMP which activates NLRP3 in alveolar macrophages leading to the release of IL-1β in mice infected with IAV [112]. This finding is significant as the NLRP3 inflammasome has been shown to have a central role in protective innate immune responses to IAVs which limits pathology and decreases mortality [113,114]. This is further reinforced by a study from Niu and colleagues which examined mice with a NLRP3 mutation (NLRP3^R258W^) that results in hyperactivation of the inflammasome [115]. These mice were found to be highly resistant to IAV infection as measured by decreased weight loss, increased survival and reduced lung pathology as compared to control mice [115]. Increased resistance was dependent on IL-1β-mediated neutrophil recruitment, as neutrophil depletion completely eliminated the significant differences in viral load, weight loss, and survival rate between the two groups [115]. Overall, there is mounting evidence for specific neutrophil dependence in the production of IL-1β and activation of NLRP3, which ultimately supports a beneficial role for neutrophils in protection against influenza viruses.

### 6.2. The Negative Roles of Neutrophils during Influenza Virus Infection

Although most IAV infections result in mild to moderate disease, severe morbidity and mortality can be observed when infections lead to viral pneumonia and/or acute respiratory distress syndrome (ARDS) [116]. Neutrophils have been implicated as a contributing factor in severe disease and ARDS caused by IAVs, as excessive recruitment and activation of neutrophils, as well as NET formation, have been documented in both mice and humans [117,118,119,120,121,122,123]. Given the many direct mechanisms for pathogen killing and clearance that neutrophils possess (as reviewed in previous sections), it is perhaps easy to understand how collateral damage could be induced to surrounding tissues during an exacerbated inflammatory response. In fact, in adults with ARDS, the level of neutrophils can be used as a prognostic factor, with higher levels being associated with severe disease and mortality [97,124]. More recent studies have corroborated this observation. In 2018, Zhu and colleagues conducted a prospective study and found that patients with more severe influenza virus infections demonstrated a 6.9- fold increase in MPO-DNA complexes compared to healthy controls [118]. MPO-DNA was used to quantitate NET formation, and the increased presence of NETs in severe infections has clinical repercussions; more specifically, MPO-DNA plasma levels in H7N9 and H1N1 patients positively correlated with acute physiology and chronic health evaluation (APACHE) II scores as well as multiple organ dysfunction syndrome and mortality [118]. Similarly, Tang and colleagues utilized a network analysis of host leukocyte responses in a cohort of influenza virus-infected patients to make similar conclusions. Of all the immunological host responses examined, neutrophils showed the strongest association with severe disease [68]. Additionally, critically ill patients demonstrated excessive NET formation, neutrophil-inflammation, and delayed apoptosis [68]. The authors further extended this relationship by employing an area-under-the receiver–operator curve comparing expression of the CD177 gene (representing neutrophil activation) and mortality to find that CD177 expression is predictive of influenza-related fatality in both patient sets and that this gene is more highly expressed in nonsurvivors than survivors [68]. Collectively, these studies point to a central role for neutrophils in severe disease.

In addition to severity, the harmful effects of neutrophils are closely linked with age [94,118,123]. Similar to clinical studies [125,126,127], infection of aged mice with IAV results in increased rates of morbidity and mortality [128,129]. Mechanistic studies to dissect the molecular mechanism responsible for this increase have implicated neutrophils as a contributing factor. Kulkarni and colleagues compared whole lung neutrophil counts in the young (2–4 month old) and aged (18–22 month old) groups infected with IAV and found that the aged mice had a significant, three-fold increase in neutrophil levels on day 6 p.i as compared to young mice [130]. While neutrophils declined in both groups after day 6, they remained elevated in aged mice and were still significantly elevated at day 12 as compared to young mice [130]. Furthermore, neutrophil depletion in aged mice increased survival rates by 40% compared to control, while in young mice there was no significant difference [130]. Interestingly, intranasal DNase treatment had no impact on survival in either group of mice, suggesting that NETs are not responsible for the increase in mortality observed in aged mice [123,130]. Along similar lines, a study by Wong et al. identified reduced phagocytosis of apoptotic neutrophils by alveolar macrophage in aged influenza-infected mice [111]. Assessment of these influenza-infected aged mice revealed that neutrophil retention in the lung and MPO levels in the BAL (a marker of activation) were increased. As a whole, these data suggest that neutrophil retention and activation contribute to the lung pathology and ultimately increased morbidity and mortality in aged mice.

An important caveat to the aforementioned studies is the inability to establish causation. While it is possible that neutrophil activity is the cause of severe disease, it is also true that exacerbated neutrophil responses are simply a consequence of the immune dysregulation that accompanies severe disease. While several hypotheses have been put forth to explain the excessive recruitment of neutrophils to the lung, there is no consensus in the field about possible upstream factors [131,132]. At present, there is a paucity of treatment options for patients with ARDS, with the majority of therapies merely providing supportive care. Such ARDS therapies include, positive-end expiratory pressure, prone positioning, and in the most severe instances extracorporeal membrane oxygenation, all of which provide varying degrees of mechanical ventilation support [133]. Pharmacologic therapies, like corticosteroids, can also be given as general anti-inflammatory agents to alleviate ARDS symptoms, however as highlighted by Carlet et al., this strategy remains controversial for virus-induced ARDS [134]. Notably, standard ARDS treatments are not specific to neutrophils, stressing the need for future work to elucidate the upstream mechanisms leading to excessive neutrophil recruitment. Importantly, in light of the ongoing COVID-19 pandemic, there has been increased interest in developing neutrophil-centered treatments and at present ongoing clinical trials are evaluating the ability of DNAses and JAK inhibitors to reduce NET formation [135]. These therapies could potentially expand the limited repertoire of treatments for ARDS patients.

### 6.3. A Negligible Role for Neutrophils in Influenza Virus Infection

While the majority of studies indicate that neutrophils impact the outcome of influenza virus infection, a select body of clinical and experimental evidence suggests that neutrophils have little role in viral clearance. Patients with chronic granulomatous disease (CGD) have mutations in NADPH oxidase which render their neutrophils incapable of ROS production [136]. While these patients experience increased susceptibility to disseminated bacterial and fungal infections (most commonly *Staphylococcus aureus*, *Aspergillus* spp., and *Burkholderia cepacia*), a disproportionate predisposition to viral infection has not been observed [137]. However, it is difficult to draw conclusions from these data for influenza virus infections, as CGD patients are able to receive seasonal inactivated influenza virus vaccines and therefore generate adaptive immune responses which can compensate for deficiencies in the innate compartment.

Data from animal models suggests that the relative contributions of neutrophils to clearance of influenza virus is dependent on viral strain and the severity of infection. Tate and colleagues examined the impact of neutrophil depletion on mice infected with avirulent strains as well as strains of intermediate and high virulence. Neutrophils were found to have a protective role for infection with intermediate and highly virulent strains, as increased morbidity and mortality was observed in neutrophil depleted as compared to controls [138]. However, neutrophil depletion had no impact on viral replication or morbidity in mice infected with an avirulent influenza virus strain [138]. Along similar lines, using CXCR2^-/-^ mice, Wareing et al. demonstrated that for a sublethal influenza virus infection, absence of the chemokine receptor decreased neutrophil recruitment to the lung but did not impact viral titers or clearance kinetics [139]. Taken together these data emphasize differential contributions of neutrophils in protection from diverse pathogens and suggest that in specific circumstances neutrophils may be dispensable for mild self-limiting influenza virus infections.

## 7. Implications and Considerations in HA-Targeted Universal Vaccine Design

The development of universal influenza virus vaccine strategies in the last decade has focused intensely on the generation of antibodies that bind the subdominant HA-stalk domain. Not only is there robust evidence that these antibodies can achieve broad protection in several animal models [140,141,142,143,144], the mechanisms through which these antibodies achieve protection is fundamentally different than HA-head specific antibodies, providing an opportunity to leverage the antibody-mediated effector functions of neutrophils in vaccine design. The level and antibody isotype of bnAbs are important considerations in moving candidate vaccines forward. Optimal protection of broadly-reactive influenza antibodies and HA-stalk specific antibodies has been shown to rely on Fc-FcR engagement [45,89], yet these studies show a clear dose-dependent effect. High levels of bnAbs can achieve protection independent of FcγR, while lower levels display an FcγR requirement [45,89]. Therefore, the levels of bnAbs and predominant isotypes elicited by a particular vaccine strategy will dictate the prevalence of effector mechanisms, with lower titers favoring Fc-mediated effector functions. In these situations, viral clearance through mechanisms such as ADCP will be of particular significance.

The vast majority of work surrounding bnAbs has focused on IgG, although secretory IgA is the most abundant isotype at mucosal surfaces including the respiratory tract. There is a paucity of data surrounding the contributions of broadly neutralizing IgA antibodies to the protection elicited by universal candidate vaccines. This is due, in part, to the difficulty in accessing and obtaining mucosal IgA samples from humans and also because the heavily relied upon mouse model lacks a FcαRI equivalent. In an in vitro setting IgA:influenza virus immune complexes have been shown to potentiate NETosis, and stalk-specific mAbs were found to be the main contributors to this process [60]. Moreover, IAV virus particles were trapped and inactivated in NETs [60]. In the context of polyclonal vaccine-induced responses, where viral replication is limited through multiple mechanisms, low levels of NET formation are likely to be protective rather than pathogenic; however, further studies are needed to test this directly. In summary, our knowledge surrounding the contributions of neutrophils to the resolution of viral infections has expanded tremendously in the past decade paving the way for rational vaccine strategies that seek to maximize their effector functions.

## 8. Conclusions

In the context of viral infections, neutrophils have traditionally been overlooked and underappreciated as an effector cell population. However, it is increasingly evident that neutrophils are critical not only in first-line innate defenses, but they also play a significant role in shaping the adaptive responses to viruses. Here, we outline an expanded role for this cell population in immunity against IAVs, given their diverse and unique effector mechanisms. Work in mouse models of influenza virus infection has identified a protective role for neutrophils in modulating T-cell responses and activating NLRP3 [106,107,112,115]. Similarly, in vitro work has demonstrated that ADCP and NETosis are elicited by influenza-specific antibodies, which may also contribute to pathogen clearance in vivo [44,60]. Yet, excessive and exacerbated neutrophilic responses are observed in severe disease and ARDS [114,115,116,117,118,119,120]. Further work is needed to understand the upstream influences that impact the magnitude of the neutrophil response and ultimately dictate protection versus pathology. This knowledge has direct therapeutic applications and will more clearly define the role of neutrophils in antiviral defense.

## Figures and Tables

**Figure 1 vaccines-09-00597-f001:**
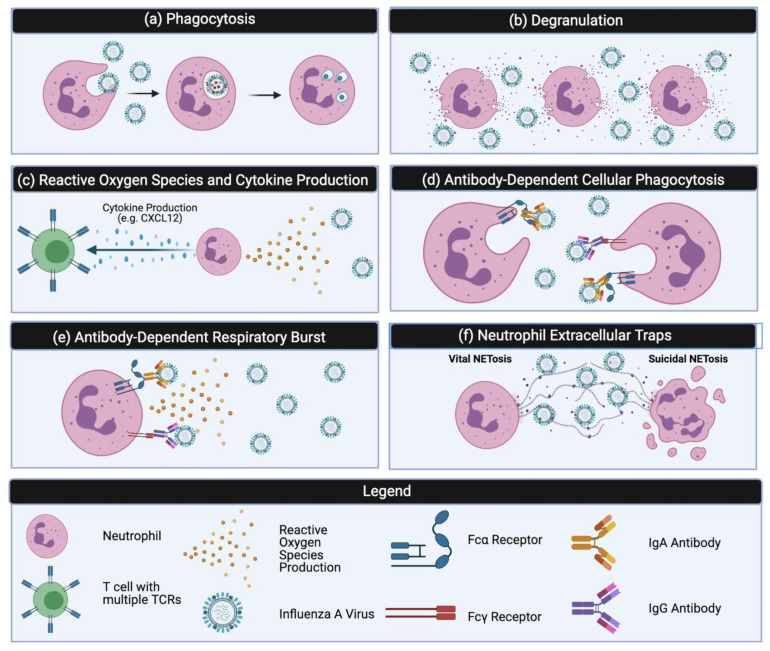
Schematic of several well-described neutrophil effector functions. (**a**) Phagocytosis is a complex process which involves the internalization and subsequent elimination of pathogens after formation of the phagosome (**b**) Degranulation involves the release of internal granules containing enzymes and cytotoxic chemicals in a controlled fashion upon inflammation or pathogen recognition. (**c**) Reactive oxygen species are generated by NADPH oxidase complex and can be released intra or extracellularly. Cytokines consist of small, secreted proteins that have a specific effect on cell–cell communication and interaction (**d**) Antibody-dependent cellular phagocytosis occurs after antigen-specific Fc-FcR interactions resulting in internalization (**e**) Antibody-dependent respiratory bursts occur after antigen-specific Fc-FcR interactions to immune complexes, resulting in ROS production. (**f**) Neutrophil extracellular traps consist of extracellular DNA fibers released due to chromatin decondensation and function to bind and immobilize pathogens. Suicidal NETosis results in cell death, while in vital NETosis the outer membrane remains intact allowing the neutrophil to continue to function. All figure panels were created with Biorender.com (accessed on 28 May 2021).

## Data Availability

Not applicable.

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
