# Peer review of "Neutrophils and Influenza: A Thin Line between Helpful and Harmful"

_vaccines, 2021, doi:10.3390/vaccines9060597_

Round 1

Reviewer 1 Report

George et al present a timely review of the role of neutrophils with a focus on their role in IAV.  Of note is the mindful comparison of murine and human responses which is much appreciated.  I feel this work is relevant and useful for publication.

Author Response

We thank the reviewer for their positive feedback and appraisal of our work.

Reviewer 2 Report

Neutrophils are an essential component of the innate immune response, and they play an important role in modulating the immune response to viral infections. However, it is unclear whether neutrophils are beneficial to the host or not during the viral infections. The authors of this manuscript summarized the current understanding of neutrophils and discussed neutrophil effector functions as well as neutrophils' contentious role in influenza virus infections.

The manuscript is well-written and summarizes recent advances in neutrophil function and potential roles during influenza A virus infection. The questions about the role of neutrophils during influenza virus infection were raised accordingly.

A recent publication provided a comprehensive review of neutrophil function (Johansson, C., Kirsebom, F.C.M. Neutrophils in respiratory viral infections. Mucosal Immunol (2021)). It would be preferable if the authors could condense their overview of neutrophils and focus on role of neutrophils in influenza virus infection.

Author Response

We thank the reviewer for alerting us to this excellent resource. We have reduced and condensed text under Section 3, “Classical Neutrophil Effector Functions” and also the subsections 4.2 “Antibody-Dependent Respiratory Burst” and  4.3 “Neutrophil Extracellular Traps”. Section 4.1 “Antibody-Dependent Phagocytosis” is very influenza-specific and thus was not altered.

Reviewer 3 Report

The review by George et al discussing the role of neutrophils in pathogen defense is comprehensive, informative, and very well written.  

One major concern is the complete absence of a discussion of the literature that supports that neutrophils have a negligent or minimal role in contributing to the elimination of viral infections such as influenza.  A number of citations comparable to those that suggest neutrophils contribute to protection are published.  In order to avoid bias and balance the discussion of the role of neutrophils in clearance of IAV infection, these should be included. 

Minor comments:

  1.  Figure 1 section F depicting neutrophil extracellular traps could be altered to include vital and suicidal NETs as they are both discussed at length within the text.
  2. Line 283: "Antigens" such as ovalbumin do not need inhospitable environments.  The term pathogen is more appropriate here. 
  3. Line 297: "that IGA-opsonized elicit" seems to be missing a word. Please clarify the intent. 
  4. Line 415 contains a duplicate period. 

Author Response

One major concern is the complete absence of a discussion of the literature that supports that neutrophils have a negligent or minimal role in contributing to the elimination of viral infections such as influenza.

  • We thank the reviewer for this thoughtful suggestion. We have included an additional section (6.3) entitled “A negligible role for neutrophils in influenza virus infections” to address this concern. Here we discuss clinical data related to patients with deficits in the neutrophil compartment such as chronic granulomatous disease, as well a select number of mouse studies which have indicated a minimal role for neutrophil contributions to influenza viral infection. 

Minor comments:

  1.  Figure 1 section F depicting neutrophil extracellular traps could be altered to include vital and suicidal NETs as they are both discussed at length within the text.
  • We appreciate the constructive feedback and close attention to detail. The figure has been modified as suggested and the figure legend has also been revised accordingly. 
  1. Line 283: "Antigens" such as ovalbumin do not need inhospitable environments.  The term pathogen is more appropriate here. 
  • We agree the term pathogens is more appropriate and have modified the text.
  1. Line 297: "that IGA-opsonized elicit" seems to be missing a word. Please clarify the intent. 
  • The word bacteria has been added to clarify the results of this study.
  1. Line 415 contains a duplicate period
  • The text has been modified to remove the duplication.

Reviewer 4 Report

The manuscript number 121957 entitled “Neutrophils and influenza: a thin line between helpful and harmful” presents an overview of the classical neutrophil effector functions and summarize recent advancements in the less well-understood antibody-dependent effector functions of neutrophils. It also focus on the involvement of neutrophils during influenza A virus infections to evaluate the beneficial and potentially detrimental role of these cells in influenza disease progression. Finally, a discussion of the possible roles neutrophils may play in the development of an universal influenza vaccine is also considered. The manuscript present valuable information, especially at this moment facing to the COVID-19 pandemic situation and the side effects induced by this disease. However some details should be improved before consideration for publication. 
-    In the abstract section, please rewrite the end of first sentence or the beginning of the second one in order to avoid the words repetition.
-    Please be more specific in the end of the abstract regarding the indication of the subjects that will be highlighted during the Review. 
-    The designation of certain pathogens and virus family names, such as Escherichia coli and respective abbreviation in line 98, Staphylococcus aureus in line269, Orthomyxoviridae line 348 should be italicized. 
-    The topic 3.3 Reactive Oxygen Species is not clear in the text the relation of Neutrophils and the induction of ROS, please clarify.
-    Please improve the sentence in line 384 in order to avoid the end with “to name a few.”
-    An improvement in the discussion of the topic 6.2 “The negative roles of neutrophils during influenza virus infection” should be considered. Regarding the recent knowledge in the COVID-19 related to the role of neutrophils induced by the infection of the respective virus (and consequently produced cytokines) and the caused adverse effects, there is some kind of therapeutic procedure that can mediate the negative roles, directly or indirectly, of these immune cells? Can be suggested the application of some kind of antiviral drugs? Or antibody blockers? What it has been studied in this field? 

Author Response

In the abstract section, please rewrite the end of first sentence or the beginning of the second one in order to avoid the words repetition.

  • The beginning of the second sentence has been edited to “annual” instead of “each year”

Please be more specific in the end of the abstract regarding the indication of the subjects that will be highlighted during the Review. 

  • The last two sentences of the abstract have been modified to address this concern.

The designation of certain pathogens and virus family names, such as Escherichia coli and respective abbreviation in line 98, Staphylococcus aureus in line269, Orthomyxoviridae line 348 should be italicized. 

  • The text has been edited to italicize the terms in the aforementioned lines.

The topic 3.3 Reactive Oxygen Species is not clear in the text the relation of Neutrophils and the induction of ROS, please clarify.

  • This section has been revised to clarify how stimulated neutrophils lead to the production of ROS.

Please improve the sentence in line 384 in order to avoid the end with “to name a few.”

  • The text has been modified to improve clarity.

An improvement in the discussion of the topic 6.2 “The negative roles of neutrophils during influenza virus infection” should be considered. Regarding the recent knowledge in the COVID-19 related to the role of neutrophils induced by the infection of the respective virus (and consequently produced cytokines) and the caused adverse effects, there is some kind of therapeutic procedure that can mediate the negative roles, directly or indirectly, of these immune cells? Can be suggested the application of some kind of antiviral drugs? Or antibody blockers? What it has been studied in this field? 

  • We thank the reviewer for this suggestion and have expanded the discussion of Section 6.2 to include treatment options for ARDS. At present, current ARDS treatment options are not neutrophil specific, however there are ongoing clinical trials to evaluate therapies to inhibit NET formation.